# *dNAGLU* Extends Life Span and Promotes Fitness and Stress Resistance in *Drosophila*

**DOI:** 10.3390/ijms232214433

**Published:** 2022-11-20

**Authors:** Rubing Xue, Ke Yang, Fuhui Xiao, Liping Yang, Guijun Chen, Yongxuan Li, Yunshuang Ye, Kangning Chen, Sheryl T. Smith, Gonghua Li, Qingpeng Kong, Jumin Zhou

**Affiliations:** 1Key Laboratory of Animal Models and Human Disease Mechanisms of Chinese Academy of Sciences/Key Laboratory of Healthy Aging Research of Yunnan Province, Kunming Key Laboratory of Healthy Aging Study, Kunming Institute of Zoology, Kunming 650223, China; 2Kunming College of Life Science, University of Chinese Academy of Sciences, Beijing 100049, China; 3State Key Laboratory of Genetic Resources and Evolution/Key Laboratory of Healthy Aging Research of Yunnan Province, Kunming Key Laboratory of Healthy Aging Study, Kunming Institute of Zoology, Chinese Academy of Sciences, Kunming 650201, China; 4Biology Department, Arcadia University, Glenside, PA 19038, USA; 5KIZ/CUHK Joint Laboratory of Bioresources and Molecular Research in Common Diseases, Kunming 650223, China

**Keywords:** *dNAGLU*, *NAGLU*, longevity, healthy aging, Alzheimer’s disease, Aβ42

## Abstract

To identify new factors that promote longevity and healthy aging, we studied *Drosophila CG13397*, an ortholog of the human *NAGLU* gene, a lysosomal enzyme overexpressed in centenarians. We found that the overexpression of *CG13397* (*dNAGLU*) ubiquitously, or tissue specifically, in the nervous system or fat body could extend fly life span. It also extended the life span of flies overexpressing human Aβ42, in a *Drosophila* Alzheimer’s disease (AD) model. To investigate whether *dNAGLU* could influence health span, we analyzed the effect of its overexpression on AD flies and found that it improved the climbing ability and stress resistance, including desiccation and hunger, suggesting that *dNAGLU* improved fly health span. We found that the deposition of Aβ42 in the mushroom body, which is the fly central nervous system, was reduced, and the lysosomal activity in the intestine was increased in *dNAGLU* over-expressing flies. When *NAGLU* was overexpressed in human U251-APP cells, which expresses a mutant form of the Aβ-precursor protein (APP), APP-p.M671L, these cells exhibited stronger lysosomal activity and and enhanced expression of lysosomal pathway genes. The concentration of Aβ42 in the cell supernatant was reduced, and the growth arrest caused by APP expression was reversed, suggesting that *NAGLU* could play a wider role beyond its catalytic activity to enhance lysosomal activity. These results also suggest that *NAGLU* overexpression could be explored to promote healthy aging and to prevent the onset of neurodegenerative diseases, including AD.

## 1. Introduction

The effects of aging are complex and diverse and aging causes gradual loss and degeneration of our body’s constituent substances, tissue structure integrity, and physical and cognitive functions. It is an unavoidable risk factor for major human diseases, including cancer and cardiovascular and neurodegenerative diseases [1]. Alzheimer’s disease (AD) is the most common age-related disease [2,3], characterized by the accumulation of intracellular neurofibrillary tangles composed of hyperphosphorylated tau protein and the extracellular aggregation of β-amyloid (Aβ) plaques [4,5,6]. Excessive Aβ production and insufficient Aβ clearance play a key role in the pathogenesis of AD [7,8].

Centenarians live much longer than the general population, characterized by much delayed onset or absence of aging-related diseases [9,10,11,12], making them excellent subjects to study the mechanism of human longevity and healthy aging. Our previous study found that the autophagy–lysosome pathway was upregulated in centenarians [13]. The product of one of these genes, N-acetyl-alpha-glucosaminidase (NAGLU), degrades heparan sulfate, and is an important glucosidase in the lysosomal pathway. Its deficiency leads to clinically termed mucopolysaccharidosis-type IIIB (MPS IIIB) [14,15], a devastating and currently incurable neurodegenerative disease, characterized by lysosomal accumulation and urinary excretion of heparan sulfate, and causes progressive cognitive decline and behavioral difficulties [16,17].

Model organisms, such as nematodes [18,19], fruit flies [20,21], and rodents [22,23], are extensively used to study mechanisms of aging and longevity. Seventy percent of human disease-related genes are conserved between flies and humans, and given its short life span, *Drosophila* is well suited to study whether centenarian overexpressed genes have the potential to extend life span or health span. *CG13397*, the *Drosophila* homologue of *NAGLU*, or *dNAGLU,* is widely expressed in the fat body, the CNS, and the heart of flies [24]. Herein, we used geneswitch (GS), a modified Gal4/UAS inducible expression system, to induce *dNAGLU* overexpression in specific tissues, and its transcriptional activity within the target tissues depends on the presence of the activator mifepristone (RU486) [25]. The GS-Gal4 flies were used to express steroid-activated GAL4 in the whole body. Then, we investigated whether and how *dNAGLU* affected the fly life span and health span by overexpressing it in wild type (wt) flies and in the *Drosophila* model of AD, which expresses human Aβ42 and GS-Gal4 simultaneously [26,27]. We found that when overexpressed either ubiquitously or tissue specifically in the fat body and nervous system, *dNAGLU* extended the life span in wt flies. In AD flies overexpressing human Aβ42, *dNAGLU* prolongs both life span and health span, demonstrated by improved climbing ability as well as resistance to stress, including that induced by desiccation and hunger. The overexpression increased lysosomal activity and reduced Aβ42 deposition in the fly CNS. The activity of *dNAGLU* was confirmed by overexpressing *NAGLU* in human U251-APP cells, which resulted in stronger lysosomal activity in these cells and a reduction in Aβ42 in the cell supernatant. Thus, *NAGLU* could extend life span and health span by improving lysosomal function.

## 2. Results

### 2.1. NAGLU Is Upregulated in Centenarians

We studied a group of 117 subjects from centenarian families, consisting of 76 centenarians (CENs) and 41 spouses of centenarian children (F1SPs, considered as younger adult controls) to analyze the differentially expressed genes in centenarians’ blood samples and found that *NAGLU* was upregulated in CENs (Figure 1A, *p* = 0.0002), while its expression patterns in controls did not show age-related changes (Figure 1B, *p* = 0.5768). We also analyzed the dataset of genome-wide RNA-seq profiles of human dermal fibroblasts from 133 people aged 1 to 94 years old [28] and found that *NAGLU* expression had no correlation with age in the general population (Figure 1C, *p* = 0.4064), suggesting that the overexpression in centenarians is unlikely the consequences of extreme aging.

### 2.2. Overexpression of dNAGLU Extends Drosophila Life Span

To study the functional consequences of overexpressing *NAGLU*, we cloned its *Drosophila* homologue *dNAGLU* and used the Gal4-UAS system to induce *dNAGLU* overexpression by supplementing with RU486. We tested the effects of RU486 on lifespan and fecundity and found that 200 μM RU486 had no significant effect on lifespan and fecundity of wt flies (Appendix A). Therefore, we used 200μM RU486 to induce *dNAGLU* overexpression. We overexpressed *dNAGLU* ubiquitously and observed its effects on *Drosophila* lifespan. The induced mRNA level of *dNAGLU* was confirmed by qRT-PCR to have an increased expression of 30.43%, *p* = 0.0062 (Figure 2A) compared to uninduced *dNAGLU*, and the expression of 3HA-dNAGLU protein was detected by Western blot (Appendix A). Statistical analysis of lifespan showed that the overexpression of *dNAGLU* prolonged the lifespan of *Drosophila*, and mean lifespan was increased by up to 3.64% (Figure 2B, *p* = 0.0004). When *dNAGLU* was knocked down in the *Drosophila* whole body, the lifespan was significantly decreased in comparison with the control group (Appendix A).

The fat body plays an important role in intermediary metabolism and is also an endocrine organ that produces a variety of antimicrobial peptides involved in the humoral immunity of *Drosophila* [29]. In contrast, the fly central nervous system regulates homeostasis and adapts to environmental changes throughout the organism and is also a target for aging and anti-aging interventions [30]. As *dNAGLU* is expressed at high levels in the fat body and mushroom body [24], we hypothesize that these tissues are where *dNAGLU* may exert its main function. We, therefore, induced *dNAGLU* overexpression in the fat body and mushroom body and detected the overexpressed *dNAGLU* mRNA in the fat body to be 41.52% (*p* = 0.0002) higher than that in the control group (Figure 2C), and the expression of 3HA-dNAGLU protein was detected by Western blot (Appendix A). Statistical analysis of lifespan showed that the overexpression of *dNAGLU* in the fat body prolonged the lifespan of *Drosophila*, and mean lifespan increased by 2.19% (Figure 2D, *p* = 0.0008). When *dNAGLU* was knocked down in the *Drosophila* fat body, the lifespan was significantly decreased in comparison with the control group (Appendix A). In the CNS, *dNAGLU* mRNA expression was higher than the control group (Figure 2E), with an increase of 177.9% (*p* = 0.0010). Additionally, the overexpression of *dNAGLU* in the nervous system prolonged the lifespan of *Drosophila*, and mean lifespan increased by 2.55% (Figure 2F, *p* = 0.0195). These results are stable after repeated verification (Appendix A). These analyses demonstrate that *dNAGLU* overexpression prolongs lifespan and suggest that it may exert its function mainly in the fat body and nervous system.

### 2.3. Overexpression of dNAGLU Extends Life Span and Health Span in the Drosophila AD Model

The up-regulation of autophagy and lysosomal signals in centenarians suggests that these pathways may have a role preventing the accumulation of molecules implicated in neurodegenerative diseases, and, therefore, reducing the risk of these diseases in these long-lived individuals. We speculated that genes in this pathway may exert a stronger effect in flies expressing human Aβ42, a *Drosophila* AD model, than in normal flies and subsequently studied how *dNAGLU* may function in the *Drosophila* AD model.

*GS-Gal4>Aβ42* flies are homozygous and carry both GS-Gal4 and UAS-Aβ42. Only when RU486 is present could *GS-Gal4>Aβ42* flies induce Aβ42 expression in the whole body. In the control group, the F1 generation obtained by crossing of *GS-Gal4>Aβ42* flies and *w^1118^* flies were fed sucrose–yeast (SY) food with 200 μM RU486 to induce ubiquitous Aβ42 expression (i.e., *GS-Gal4>Aβ42 >*+ +RU flies, this is the *Drosophila* AD model). While in the experimental group, the F1 generation obtained by crossing of *GS-Gal4>Aβ42* flies and *dNAGLU* transgenic flies were fed SY food with 200 μM RU486 to induce both the expression of Aβ42 and the overexpression of *dNAGLU* (i.e., *GS-Gal4>Aβ42>dNAGLU* +RU flies, in other words, we induced *dNAGLU* overexpression in *Drosophila* AD model). We fed flies different foods at different stages (Figure 3A) to measure the effects of *dNAGLU* overexpression on the lifespan and health span. Again, the mRNA of *dNAGLU* in induced *Drosophila* was found to be 42.91% (Figure 3B, *p* = 0.0018) higher than uninduced group, and the expression of 3HA-dNAGLU protein was detected by Western blot (Appendix A). The overexpression of *dNAGLU* in AD *Drosophila* significantly prolonged the lifespan in these flies, and themean lifespan was increased by up to 7.01% (Figure 3C, *p* = 0.0004).

A previous study showed that *Drosophila* exhibited severe age-related deterioration in skeletal muscle function, such as reduced flight time and impaired climbing ability [31]. Studies in yeast have confirmed that activation of an autophagy-lysosomal pathway is critical for maintaining cell viability during extreme nutrient conditions, such as starvation and desiccation [32], and that lysosomal activity decreases with age in most tissues and in most organisms [33]. We, therefore, assessed the fitness of these flies by subjecting 20-day-old flies to climbing test, desiccation, starvation, and H_2_O_2_ treatment to evaluate how *dNAGLU* impacts these functions.

We found that the climbing ability increased in *dNAGLU* overexpressing flies compared to controls, with an increase of 23.44%, *p* = 0.0158 (Figure 3D). When stress resistance was analyzed, we found that the flies overexpressing *dNAGLU* have better tolerance to starvation (Figure 3E, *p* < 0.0001) and desiccation (Figure 3F, *p* = 0.0018) than control flies, and the mean survival time was increased by 14.10% and 9.25%, respectively. However, no significant change was seen in tolerance to H_2_O_2_ treatment, although the *dNAGLU* overexpression group showed a tendency of increased survival (Figure 3G, *p* = 0.1850). These results are stable after repeated verification (Appendix A).

In the negative control group, F1 flies from the above crosses were fed with normal SY food without RU486 but supplemented with the same dose of ethanol (i.e., *GS-Gal4>Aβ42>+* −RU flies and *GS-Gal4>Aβ42>dNAGLU* −RU flies), which induced neither the expression of Aβ42 nor the overexpression of *dNAGLU*, and no significant difference was observed in the lifespan among these flies. Similarly, their climbing ability and tolerance to desiccation, starvation, and H_2_O_2_ showed no significant differences (Appendix A). Thus, the changes we observed were due to *dNAGLU* overexpression. These results demonstrated that overexpression of *dNAGLU* in AD flies significantly extends the lifespan and improves climbing ability and stress resistance, including resistance to desiccation and hunger.

### 2.4. Overexpression of dNAGLU Reduces Aβ42 Deposition in the Mushroom Body in AD Drosophila

Since lysosomal dysfunction is a common abnormality in neurodegenerative diseases, which leads to impaired Aβ clearance [34], we attempted to determine whether *dNAGLU* could improve fly fitness and stress resistance by promoting Aβ clearance. We dissected the mushroom bodies of 50-day-old flies for cryo-sectioning and performed immunofluorescence staining to observe the deposition of Aβ42 in the brain tissue of these flies. We used *GS-Gal4>Aβ42>+* −RU flies (Figure 4A), which did not induce Aβ42 expression, as the negative control, and used *GS-Gal4>Aβ42>*+ +RU flies (Figure 4B), the *Drosophila* AD model, which induces ubiquitous Aβ42 expression, as the control. We used *GS-Gal4>Aβ42>dNAGLU* +RU flies (Figure 4C), which induce the overexpression of *dNAGLU* in the *Drosophila* AD model, as the experimental group.

We observed a dramatic decrease of Aβ42 fluorescence intensity when *dNAGLU* was overexpressed in AD flies (compare Figure 4B,C). This decrease (52.07% ) was significant when 15 brain slices were analyzed for fluorescence intensity (Figure 4D, *p* = 0.0024). In comparison, no significant difference in the mean fluorescence intensity of DAPI was seen in these tissues (Figure 4E, *p* = 0.7780). Therefore, overexpression of *dNAGLU* can significantly reduce the Aβ42 deposition in the AD *Drosophila* model.

### 2.5. Overexpression of dNAGLU in AD Drosophila Enhances the Intestinal Lysosomal Activity

As lysosomal dysfunction is associated with aging, lysosomal activity decreases with age in most tissues and organisms [33,35]; therefore, detecting the activity of lysosomes is also an important indicator for evaluating the biological health of an organism. Lyso Tracker Red can be used to label and monitor acidic lysosomes in living cells. To reveal whether *dNAGLU* overexpression increased lysosomal activity, we performed lysosomal probe staining in dissected fly intestines. Here, we used *GS-Gal4*>*Aβ42*>+ +RU flies as the control group (Figure 5A) and used *GS-Gal4*>*Aβ42*>*dNAGLU* +RU flies as the experimental group (Figure 5B). We found that the overexpression of *dNAGLU* enhanced the fluorescence intensity of gut lysosomes by 2.48-fold (Figure 5C, *p* = 0.0024), while there was no significant difference in the mean fluorescence intensity of DAPI (Figure 5D, *p* = 0.6908) in the gut of these 50-day-old AD flies, indicating that the overexpression of *dNAGLU* resulted in stronger lysosomal activity and promoted lysosomal biogenesis in these flies.

### 2.6. Overexpression of NAGLU Reduces Aβ42 Concentration in Human U251-APP Cells

In order to confirm that *NAGLU* and *dNAGLU* are functionally conserved and can indeed degrade Aβ42 in both fly and mammalian cells, we tested *NAGLU* overexpression in a human glioma cell line, U251-APP cells [36], and detected a 105-fold increase in *NAGLU* mRNA expression compared to control (Figure 6A, *p* = 0.0001). We also detected the expression of Flag-NAGLU and APP by Western blot and found that when *NAGLU* was overexpressed in U251-APP cells, there was no significant change in the expression of APP (Appendix A). So *NAGLU* overexpression did not affect the expression of APP in the original cell system. We then used ELISA to detect the concentration of Aβ42 in cell supernatant and found that the overexpression of *NAGLU* led to the recovery of cell proliferation (Figure 6B), and the Aβ42 in cell supernatant was reduced by 10.41% (Figure 6C, *p* = 0.0422). These results demonstrated that when *NAGLU* was overexpressed in human U251-APP cells, the concentration of Aβ42 in the cell supernatant was reduced, and the growth arrest caused by APP expression was reversed.

When stained lysosomes of U251-APP cells with Lyso Tracker Red (Figure 6D,E) and counted the average fluorescence intensity of cells, we found that these cells had an enhanced fluorescence intensity of 2.73-fold compared to control (Figure 6F, *p* = 0.0209), while there was no significant difference in the mean fluorescence intensity of copGFP (Figure 6G, *p* = 0.8375). The tripeptidyl peptidase (TPP) and the Cts family of cysteine proteases, i.e., –CTSB, CTSD, CTSK and CTSL, which are proteolytic enzymes in lysosomes that can degrade Aβ, and TFEB (transcription factor EB)/TFE3 (transcription factor binding to IGHM enhancer 3) are master transcriptional regulators of lysosomal activity [37]. When *NAGLU* overexpressed in U251-APP cells, transcript levels of *CTSB*, *CTSD*, *CTSK*, *CTSL*, *TPP1*, *TFEB*, and *TFE3* were significantly increased, while, other enzymes that can degrade Aβ, such as matrix metalloproteinase (MMP)-2, MMP-9 and angiotensin-converting enzyme (ACE), were also increased (Figure 6H). These results indicated that when *NAGLU* is overexpressed in U251-APP cells, it exhibited strong lysosomal activity and promoted lysosomal biogenesis, and *NAGLU* promoted the degradation of Aβ42 by enhancing the lysosomal pathway.

## 3. Discussion

In this study, we investigated the effect of overexpression *Drosophila dNAGLU*, fly ortholog of human *NAGLU*, a lysosomal enzyme highly expressed in longevity individuals and found that *dNAGLU* overexpression leads to life span extension and increased fitness and stress resistance in flies. In addition, we demonstrated that *NAGLU* overexpression reduced Aβ42 deposition and enhanced lysosomal function in AD flies and in human U251-APP cells. These results suggest that the overexpressed *NAGLU* level could potentially be a positive factor for health span and life span in humans.

*NAGLU* is located in lysosomes and is an important glucosidase in the lysosomal pathway. The autophagy-lysosome pathway degrades and recycles cellular components, including toxic protein aggregates and damaged organelles [38]. Growing evidence suggests that lysosomal dysfunction plays a key role in the development of AD. Impaired clearance of Aβ is a potential mechanism of widespread sporadic AD [8]; thus, the health and survival of neurons in the brain depend on the normal lysosomal function [39]. Lysosomal dysfunction is associated with aging, and blockade of autophagic flux may result from lysosomal damage [35]. Our study described herein suggests that *NAGLU* may have a wider role, as we demonstrated here, that in AD flies, as well as in cultured cells, *NAGLU* could enhance the lysosomal activity and promote lysosomal biogenesis.

The number of lysosomes decreased during aging and lysosomal dysfunction also lead to decreased muscle mass and function [40]. Furthermore, lysosomes can recycle amino acids required for protein synthesis under extreme nutritional conditions, such as starvation and desiccation, by degrading protein products [41]. Our study demonstrates that overexpression of *dNAGLU* in AD flies improved climbing ability, as well as starvation and desiccation resistance and ultimately prolonged lifespan. Combined with our findings of enhanced lysosomal activity, we can conclude that *dNAGLU* could improve health and prolong lifespan by enhancing lysosomal function.

Previous studies suggested that *NAGLU* deficiency in humans could lead to MPS IIIB [14,15]; thus, an overexpression could at least reduce the risk for MPS IIIB. It is shown that in a mouse model of MPS IIIB, using enzyme replacement therapy by injecting modified *NAGLU* into the brain could reduce the pathological accumulation of heparan sulfate and other metabolites (e.g., glycine, β-amyloid, P-tau) to normal or near-normal levels [42]. While β-amyloid and P-tau are still the most acknowledged pathological hallmarks of AD [43], it hints that a higher level of *NAGLU* may play a role in preventing AD. Our study demonstrates that overexpression of *NAGLU* homologue *dNAGLU* in AD flies significantly reduced Aβ42 deposition in the brain, and similar results were seen in *NAGLU* over-expressing U251-APP cells. When *NAGLU* overexpressed in U251-APP cells, the concentration of Aβ42 in the cell supernatant was reduced, and the cell cycle dysregulation caused by APP expression was reversed resulting in normal cell proliferation. Moreover, our results also showed that the mRNA levels of many proteolytic enzymes that degrade Aβ were found to be upregulated in lysosomes, e.g., *CTSB*, *CTSD*, *CTSK*, *CTSL* and *TPP1*. Importantly, we observed an increase in the mRNA levels of upstream transcription regulators *TFEB* and *TFE3*. Thus, *NAGLU* may promote lysosomal biogenesis through upregulating *TFEB*/*TFE3*, which increase the expression of a number of proteolytic enzymes that degrade Aβ in lysosomes and facilitate the degradation of Aβ42 (Figure 7). The mechanism leading to upregulation of *TFEB*/*TFE3* is currently unknown, and the relationship between *NAGLU* overexpression and autophagy also needs to be explored. These are interesting questions for future studies.

Studies have reported an intracerebral gene therapy with recombinant AAV vector serotype 2/5 encoding human *NAGLU* in four children with MPS IIIB syndrome. At intermediate analysis 30 months after surgery, neurocognition was improved in all patients [44]. After 5.5 years of follow-up, good tolerance, sustained NAGLU production, and milder disease were demonstrated in the patient treated at a very early stage [45]. As MPS IIIB is also a neurodegenerative lysosomal storage disorder, the improved neurocognition is consistent with what our findings suggest: that *NAGLU* overexpression might alleviate the symptoms of AD. As *NAGLU* is significantly upregulated in centenarians, and as its deficiency led to MPS IIIB, it is essential to investigate whether its up-regulation is a result of extreme aging or a causal factor for healthy aging and longevity. Our *Drosophila* study demonstrated that *NAGLU* is a positive factor for health aging and longevity, and its high expression in centenarians is likely a reason for longevity, rather than a result of extreme aging. Our study also suggests that *NAGLU* could function as a potential drug target for clinical intervention of aging-related neurodegenerative diseases.

Taken together, we revealed that *NAGLU* could extend lifespan and health span in flies, and it enhanced β amyloid clearance, as demonstrated in AD flies and in human cells, by enhancing lysosomal pathway. Therefore, *NAGLU* may be directly or indirectly involved in β-amyloid clearance, suggesting that *NAGLU* overexpression, or administering its agonist if it exists, could treat AD in experimental animal models, and *NAGLU* could be a potential intervention target to alleviate AD symptoms in humans.

## 4. Materials and Methods

### 4.1. Drosophila Stocks and Construction of the Transgenic Flies

The construction of transgenic flies used molecular cloning technology to insert *dNAGLU* into the UAS promoter-driven vector *pUAST-attB-3HA* to generate *pUAST-attB-3HA-dNAGLU* plasmid, which was microinjected into *86F (III) 6110* transgenic strain (gift of Sun Yat-Sen University) with the attP site inserted near 86F. *dNAGLU* transgenic flies was obtained by standard transgenic. The *dNAGLU* transgenic flies were then backcrossed with *w^1118^* flies for at least 6 generations to remove the background. *GS-Gal4* flies expressed steroid-activated GAL4 in the whole body. *S_1_106-Gal4* flies expressed steroid-activated GAL4 in the fat body. The *43642* flies expressed steroid-activated GAL4 in neurons. *GS-Gal4>Aβ42* flies expressed steroid-activated GAL4 and UAS-Aβ42 simultaneously. *w^1118^* flies were wild type flies. The *51808* flies expressed dsRNA for RNAi of *dNAGLU* under UAS control. *GeneSwitch-Gal4*, *S_1_106-Gal4*, *GS-Gal4>Aβ42*, *w^1118^* (the above flies were gifts from Sichuan Agricultural University), *43642* flies, and *51808* flies were obtained from the Bloomington *Drosophila* Stock Center (http://flystocks.bio.indiana.edu, accessed on 1 October 2020).

### 4.2. Husbandry and Life Span Analysis

The larval stage of all strains of *Drosophila* were fed standard cornmeal–yeast medium (agar 7.6 g/L H_2_O, soy flour 13.5 g/L H_2_O, corn flour 92.7 g/L H_2_O, malt flour 61.8 g/L H_2_O, yeast 22.9 g/L H_2_O, syrup 206.1g/L H_2_O, propionic acid 6.4mL/L H_2_O). During the adult experimental stage, flies were fed sucrose–yeast (SY) food (yeast 100 g/L H_2_O, agar 15 g/L H_2_O, sugar 50 g/L H_2_O, propionic acid 3 mL/L H_2_O, 10% methyl nuns 30 mL/L H_2_O).

The *dNAGLU* transgenic flies were crossed with driver lines *GS-Gal4*, *S_1_106-Gal4*, and *43642* flies, respectively. In the experimental group, the F1 generation flies were fed SY food supplemented with RU486 (*w*/*v*, 200 µM; Sigma-Aldrich, St. Louis, MO, USA, CAS 84371-65-3) to induce *dNAGLU* overexpression in the whole body, fat body, and nervous system, respectively. Additionally, in the control group, F1 generation flies were fed SY food without RU486 but supplemented with the same dose of ethanol.

After hatching, the larvae were mated in standard cornmeal–yeast medium for at least 48 h, and female flies were selected for experimental research. Using *Drosophila* for lifespan statistics with 10 flies/tube, each tube contained an appropriate amount of SY food, at least 10 tubes per experimental condition. All stocks were maintained at 22 °C and 60% humidity, 12 h light, 12 h dark cycle. The flies were transferred to fresh food every 2 days, during which the number of dead and escaped flies was recorded. All lifespan data were analyzed with the log-rank test.

### 4.3. Health Test Assays

A total of 10 flies/tube were used for the test, with at least 5 tubes per experimental condition. For the desiccation test, the flies were placed in an empty tube without medium, and the number of dead flies was counted every 2 h. For the starvation test, flies were placed in tubes containing 1.5% agarose, just to provide moisture but without any other nutrients. The number of dead flies was counted every day [21]. For the H_2_O_2_ oxidative stress test, circular filter paper was placed at the bottom of the culture tube, 50 μL of 6% glucose solution containing 30% H_2_O_2_ was added, the flies were put into the tube, and the number of dead flies was counted every 2 h [46]. All survival data were analyzed with the log-rank test. For the climbing test, 6 flies/tube were used for the test, with at least 4 tubes per experimental condition. We calculated the average speed of flies climbing from the bottom of the tube to the top within 5s to evaluate the climbing ability of flies. The climbing data were analyzed by a Student *t*-test.

### 4.4. Immunofluorescence

The brains of flies were dissected for cryosections, and the brain sections were blocked with blocking buffer (1×PBS/5%BSA/0.3%Triton X-100) at room temperature for 1 h. The slides were incubated with primary antibody: β-Amyloid (CST, 8243S) in the staining buffer (1% BSA and 0.3% Triton X-100 in PBS) at 4 °C overnight and subsequently incubated with Alexa Fluor 594-conjugated secondary antibody (ThermoFisher, Waltham, MA, USA, A-11012) for 2 h at room temperature. The slides were mounted with mounting medium with DAPI (Abcam, Cambridge, UK, ab104139). Imaging was then performed with an ultra-high-resolution laser confocal microscope (Zeiss, Oberkochen, Germany, LSM880), and mean fluorescence intensity analysis was performed with Image J.

### 4.5. Autolysosome Staining

*Drosophila* guts were dissected and stained with Lyso Tracker Red (Beyotime, C1046), and mounted with mounting medium with DAPI (Abcam, Cambridge, UK, ab104139). For living cells, operations were performed according to the manufacturer’s procedures. Imaging was then performed immediately with an ultra-high-resolution laser confocal microscope (Zeiss, Oberkochen, Germany, LSM880), and mean fluorescence intensity analysis was performed with Image J.

### 4.6. qRT-PCR

RNA was extracted with RNAiso Plus (TaKaRa, Kusatsu, Japan, RNAA00250); then, RNA was reverse transcribed into cDNA with Prime Script TMRT reagent kit with gDNA Eraser (TaKaRa, Kusatsu, Japan, RR047Q), and qRT-PCR was performed using the Fast Start Universal SYBR Green Master (ROX) (Roche, Basel, Switzerland, 04913850001) and gene-specific primers. All the above experimental operations were performed according to the manufacturer’s procedures. Gene expression levels were calculated using the comparative C_T_ method. All primers used are listed in (Appendix A).

### 4.7. Western Blot

The total protein was extracted with RIPA lysis buffer (Beyotime, Haimen, China, P0013B), and the protein extracts were added to the lanes of SDS-PAGE gels and transferred to PVDF membranes, and the membranes were incubated with antibodies. The primary antibody was anti-HA monoclonal tag (Abcam, Cambridge, UK, ab1424), anti-FLAG (Abcam, Cambridge, UK, ab205606) and anti-APP (Abclonal, Woburn, MA, USA, A17911). The secondary antibody was goat anti-mouse IgG (Invitrogen, Waltham, MA, USA, 31430) and goat anti-rabbit IgG (Invitrogen, Waltham, MA, USA, 31460). The internal reference antibody was anti-tubulin (Abcam, Cambridge, UK, ab6160) and anti-β-actin (TransGen Biotech, Beijing, China, HC201). Immobilon Western Chemiluminescent HRP Substrate (Millipore, Burlington, MA, USA, WBKLS0500) was used for chemiluminescent color development.

### 4.8. NAGLU Expression Analysis

The differences of *NAGLU* expression between the CENs and F1SPs were analyzed, as described in our previous study [13]. Pearson’s correlation test was used to evaluate the association between gene expression and age through the cor.test function in the R platform.

### 4.9. Cells and Transfection

The U251 cell line, with stable expression of the APP mutant APP-p.M671L (U251-APP cells), was a gift from Yonggang Yao [36]. *NAGLU* overexpression plasmid was constructed by using the *pCDH-CMV-MCS-EF1-copGFP-T2A-Puro* lenti-viral vector and was then transduced into U251-APP cells. Stably transfected cells were selected by fluorescence screening.

### 4.10. Cell Proliferation Analysis

We used Enhanced Cell Counting Kit-8 (Beyotime, Haimen, China, C0042) to detect cell proliferation and growth activity. We detected at 12h, 36h, 60h, 84h, and 120h after *NAGLU* overexpression. The experimental operations were performed according to the manufacturer’s procedures.

### 4.11. Aβ42 ELISA Analysis

U251-APP cells supernatant was measured using commercial ELISA kits (Elabscience, Wuhan, China, E-EL-H0543c) to detect human Aβ42. The ELISA was performed to detect Aβ42 according to the manufacturer’s instructions.

## Figures and Tables

**Figure 1 ijms-23-14433-f001:**
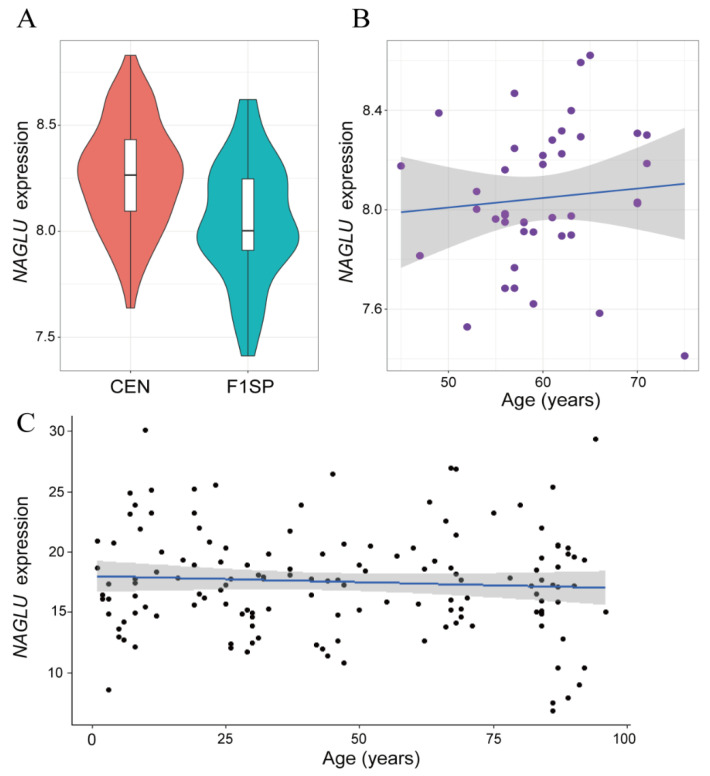
Expression patterns of *NAGLU* in CEN families and in general population: (**A**) Average expression of *NAGLU* in CENs and in F1SPs, *p* = 0.0002; (**B**,**C**) Pearson correlation test between *NAGLU* expression and age in F1SP samples, *p* = 0.5768 (**B**) and in general population, *p* = 0.4064 (**C**).

**Figure 2 ijms-23-14433-f002:**
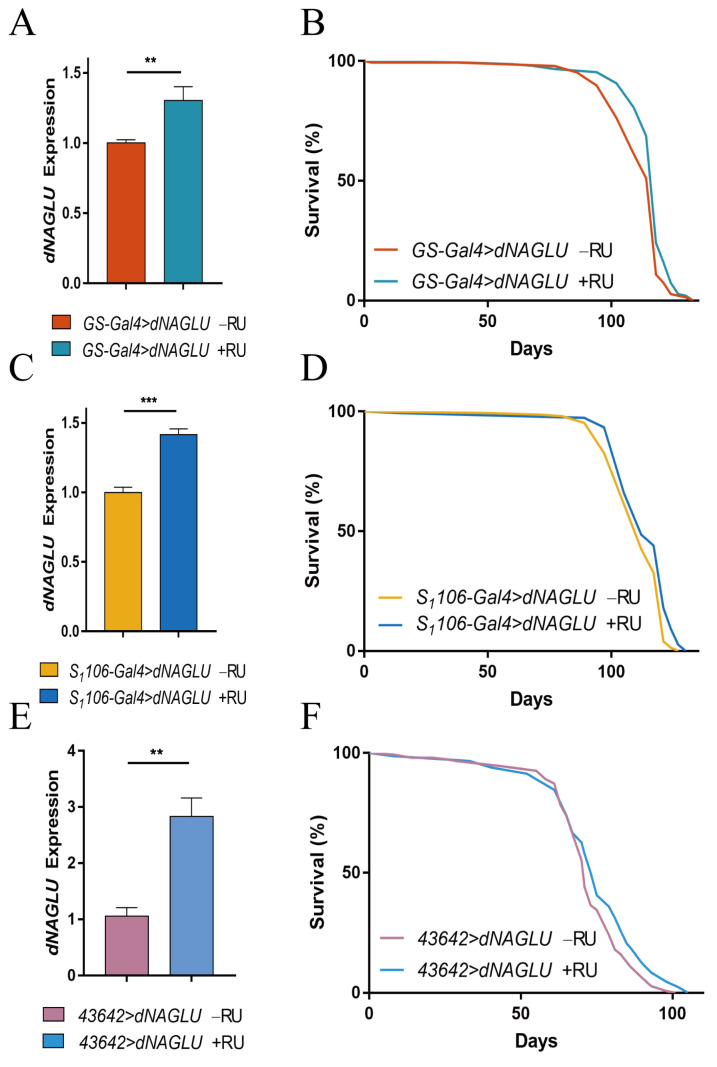
Effects of *dNAGLU* overexpression on the lifespan in *Drosophila*: (**A**) qRT-PCR validation for expression of *dNAGLU* in *Drosophila* whole body; (**B**) survival curves of *Drosophila* overexpressing *dNAGLU* in the whole body, *p* = 0.0004; (**C**) qRT-PCR validation for expression of *dNAGLU* in *Drosophila* fat body; (**D**) survival curves of *Drosophila* overexpressing *dNAGLU* in the fat body, *p* = 0.0008; (**E**) qRT-PCR validation for expression of *dNAGLU* in *Drosophila* nervous system; (**F**) survival curves of *Drosophila* overexpressing *dNAGLU* in the nervours system, *p* = 0.0195. A 5% or lower *p* value is considered to be statistically significant using Student’s *t*-test, ** *p* < 0.01, *** *p* < 0.001. All survival data were analyzed with the log-rank test.

**Figure 3 ijms-23-14433-f003:**
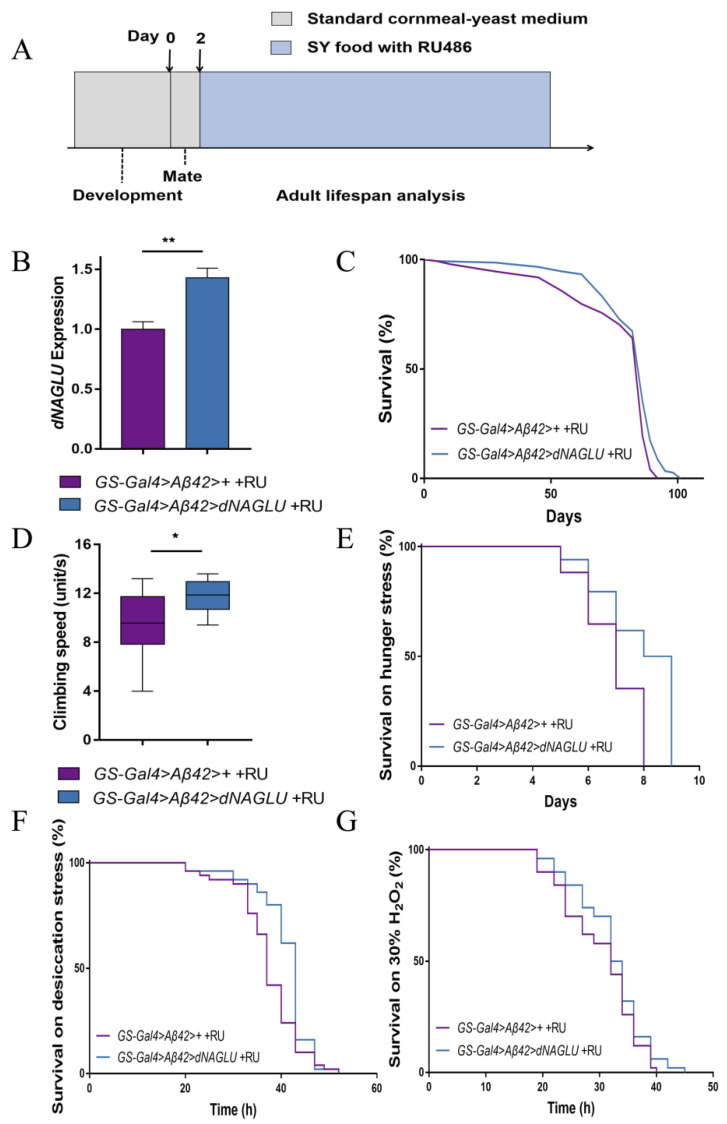
Effects of *dNAGLU* overexpression on the lifespan in *Drosophila* AD models: (**A**) Experimental timeline for feeding adult flies to experimental diets; (**B**) qRT-PCR validation for expression of *dNAGLU* in AD flies (** *p* < 0.01, *t*-test); (**C**) survival curves of *Drosophila* overexpressing *dNAGLU* in AD flies (*p* = 0.0004, log-rank test); (**D**,**E**) effects of *dNAGLU* overexpression on stress tolerance in AD flies; (**D**) climbing assay, (* *p* < 0.05, *t*-test); (**E**) hunger stress assay, (*p* < 0.0001, log-rank test); (**F**) desiccation stress assay, (*p* = 0.0018, log-rank test); (**G**) H_2_O_2_ stress assay, (*p* = 0.1850, log-rank test).

**Figure 4 ijms-23-14433-f004:**
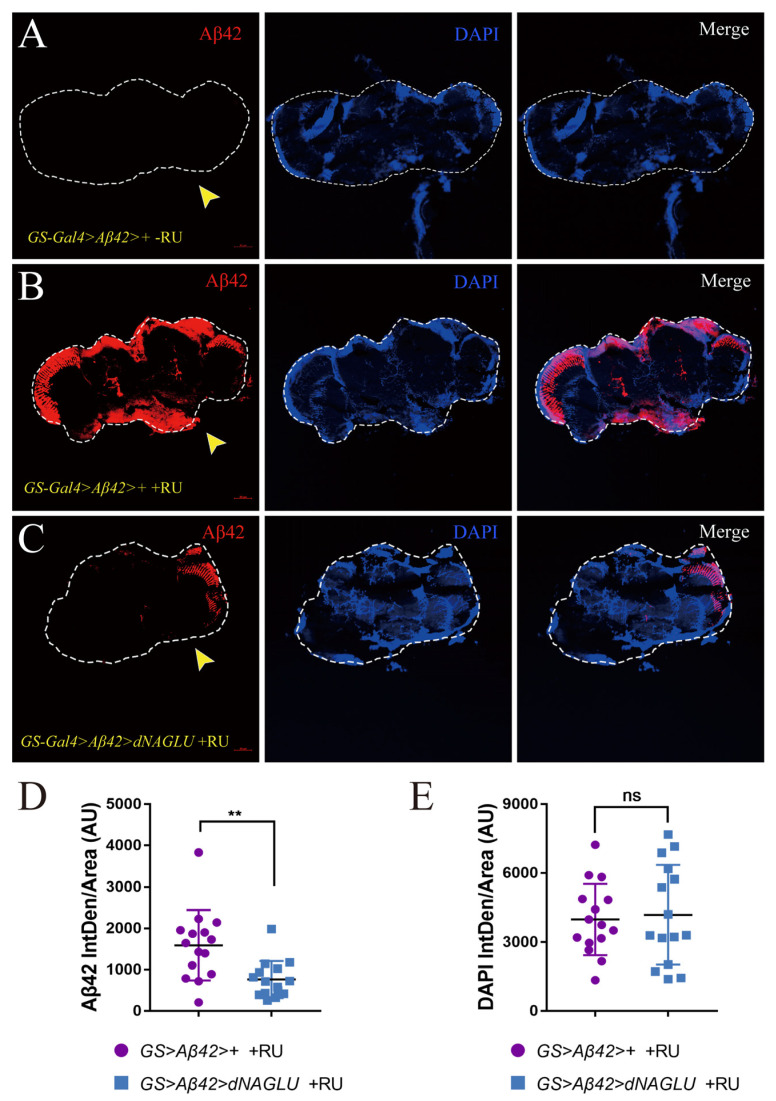
Overexpression of *dNAGLU* reduced Aβ42 deposition in the mushroom body in AD *Drosophila*: (**A**–**C**) Immunofluorescence using β-amyloid antibody in *Drosophila* brain slices; (**A**) *GS-Gal4>Aβ42>+* −RU flies; (**B**) *GS-Gal4>Aβ42>*+ +RU flies; (**C**) *GS-Gal4>Aβ42>dNAGLU* +RU flies; (**D**) the integrated density of Aβ42/area are shown as bar plots (*n* = 15, *p* = 0.0024, *t*-test); (**E**) the integrated density of DAPI/area are shown as bar plots (*n* = 15, *p* = 0.7780, *t*-test). The nuclei are stained with DAPI in blue. ** *p* < 0.01, ns represents non-significant.

**Figure 5 ijms-23-14433-f005:**
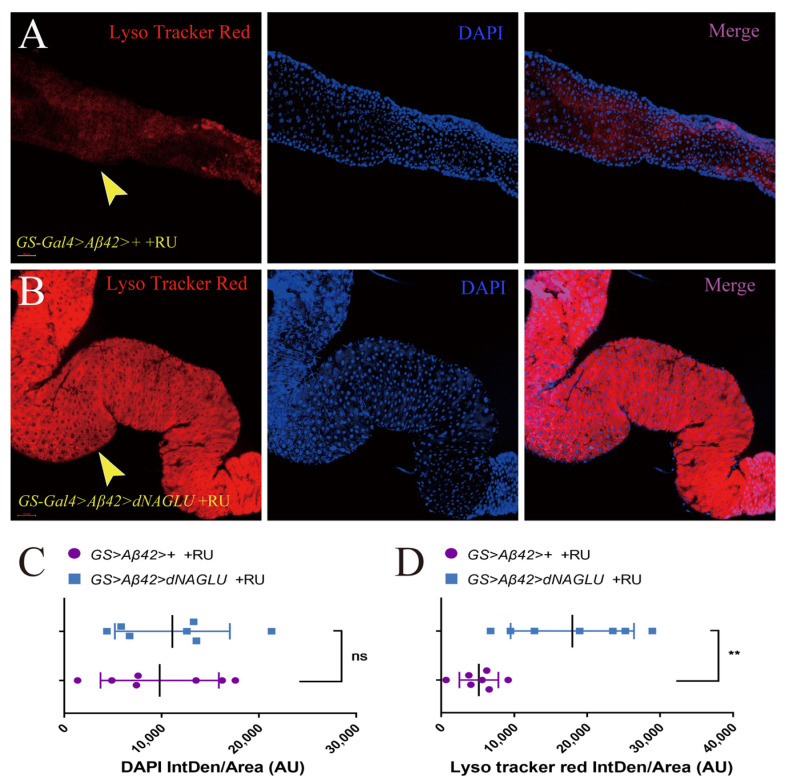
Overexpression of *dNAGLU* in AD *Drosophila* enhances intestinal lysosomal activity: (**A**,**B**) Immunofluorescence using Lyso Tracker Red in *Drosophila* guts. (**A**) *GS-Gal4>Aβ42>*+ +RU flies; (**B**) *GS-Gal4>Aβ42>dNAGLU* +RU flies; (**C**) the integrated density of Lyso Tracker Red/area are shown as bar plots (*n* = 7, *p* = 0.0024, *t*-test); (**D**) the integrated density of DAPI/area are shown as bar plots (*n* = 7, *p* = 0.6908, *t*-test). The nuclei are stained with DAPI in blue. ** *p* < 0.01, ns represents non-significant.

**Figure 6 ijms-23-14433-f006:**
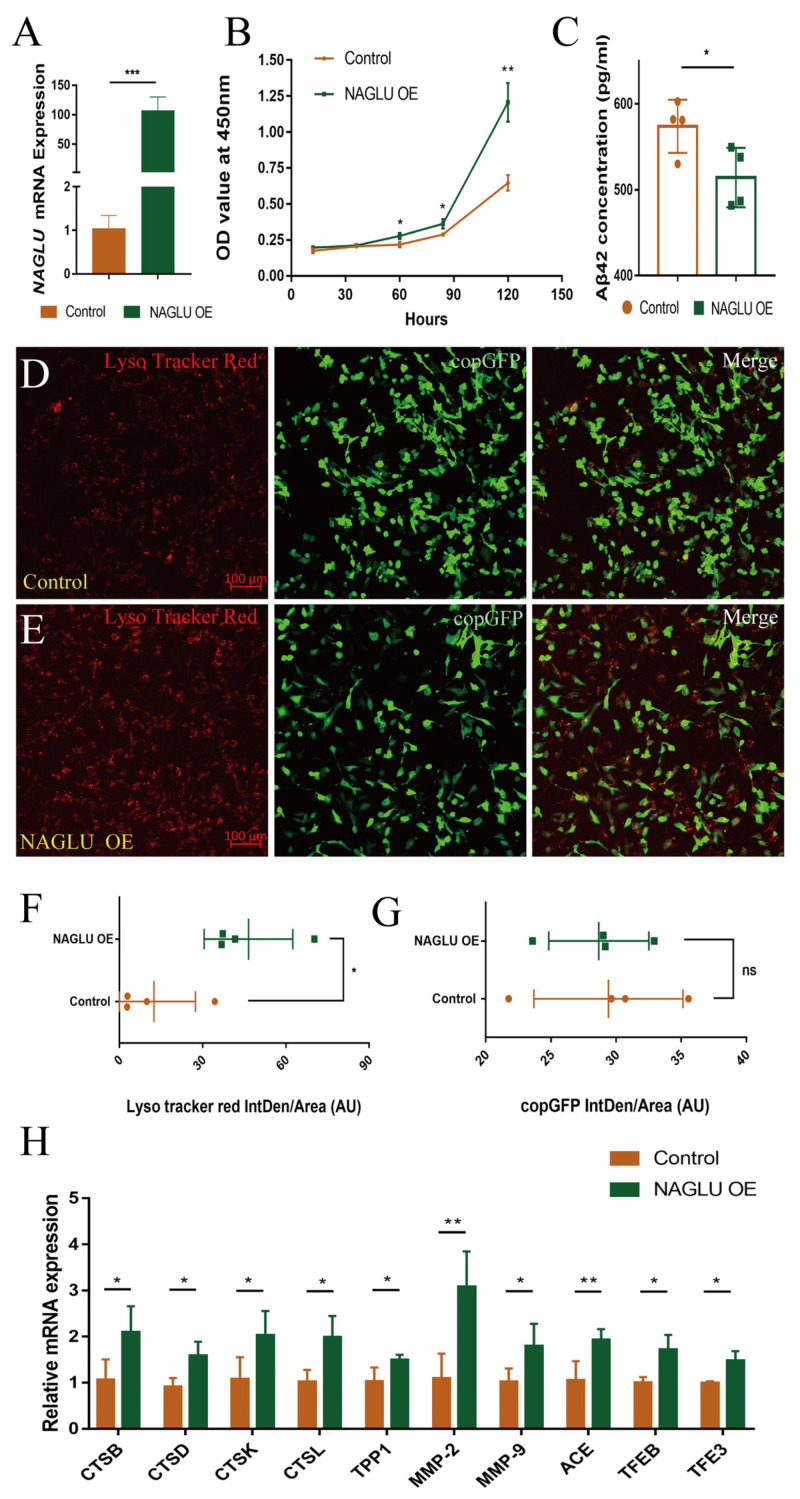
Effects of *NAGLU* overexpression in human U251-APP cells: (**A**) qRT-PCR validation for expression of *NAGLU* in U251-APP cells (*** *p* < 0.001, *t*-test); (**B**) cell proliferation curve for *NAGLU* over-expressing cells detected by cell counting assay (* *p* < 0.05, ** *p* < 0.01, *t*-test); (**C**) detection of Aβ42 concentration in cell supernatants by ELISA (* *p* < 0.05, *t*-test); (**D**,**E**) immunofluorescence using Lyso Tracker Red in cultured cells; (**D**) U251-APP cells; (**E**) *NAGLU* overexpressing U251-APP cells; (**F**) the integrated density of Lyso Tracker Red/area are shown as bar plots (*n* = 4, *p* = 0.0209, *t*-test); (**G**) the integrated density of copGFP/area are shown as bar plots (*n* = 4, *p* = 0.8375, *t*-test). * *p* < 0.05, ns represents non-significant; (**H**) relative mRNA levels of genes involved in Aβ degradation and lysosomal pathway (* *p* < 0.05, ** *p* < 0.01, *t*-test). (OE represents overexpression.)

**Figure 7 ijms-23-14433-f007:**
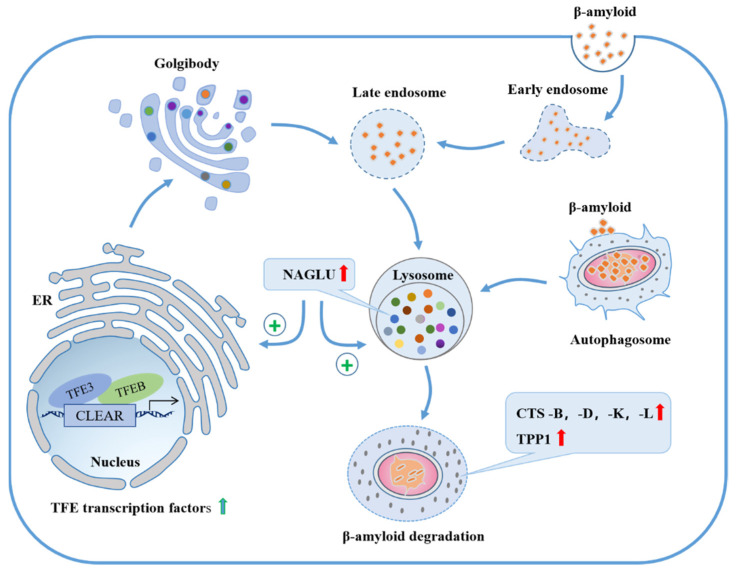
The functional model of *NAGLU*. A possible mechanism of *NAGLU* promoting the degradation of Aβ42 by enhancing the lysosomal pathway.

## Data Availability

The data presented in this study are available on request from the corresponding author.

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
