# Peer review of "dNAGLU Extends Life Span and Promotes Fitness and Stress Resistance in Drosophila"

_ijms, 2022, doi:10.3390/ijms232214433_

Round 1

Reviewer 1 Report

Xue et al. present a quite exhaustive study on the function of the Drosophila homologue of the human NAGLU gene, which they find up-regulated in centenaries. They show that over-expression of dNAGLU extends lifespan in flies and rescues phenotypes in the Alzheimer’s disease fly model. They propose that NAGLU reduces aβ deposition by improving lysosomal activity. Only at the end of the discussion, the authors mention the fact that over-expression of NAGLU cures children with MPS IIIB syndrome (which should be better explained). As a NAGLU-based therapy exists for MPS IIIB patients, what is
really missing is a discussion of the interest to use Drosophila to study NAGLU. Also, a better description of the protein function with relevant references is needed.

The manuscript requires professional English editing. Just as an example, at line 27, “overexpressed” should be replaced by “overexpressing” and a dash should be put (over-expressing). Many other mistakes are present throughout the text. Below, major and minor points that should be addressed:

Line 44. Aβ is not defined. Should be “amyloid-β (Aβ)” in the sentence before.

Line 63. The Drosophila model of AD should be explained. The manuscript is not written for a non-specialist. Everywhere, its should be Alzheimer’s, not Alzheimer. Reference 26 is
not on AD.

Line 74. Results can not be started with “previous studies”. These should go in the Introduction. Previous data can not be used to make a new figure.

Line 106. “Normally” is inappropriate. A reference should be cited.

Line 127. What is GS-Gal4? In which tissue is it expressed?

Line 128. RU486 is mentioned for the first time and no explanation is given.

Lines 127-132. This sentence is too long and incorrect. At least one reference for the AD Drosophila model should be cited.

Line 134. Figure 3 can not be started with G. This should be A.

Line 135. “Health span” is very vague. Please specify what is meant.

Line 179. GS- 178 Gal4>Aβ42>+ -RU is not clear. “+ -“ is confusing. Same for “+ +”.

Lines 178-183. This sentence is too long and incorrect.

Line 231. A conclusion should be added.

Line 280. MPS IIIB has to be explained.

Line 321. What is “ATTP site”?

Line 321. What does it mean “dNAGLU transgenic flies 321 can be obtained through mating screening.”? Have they been obtained by standard transgenesis?

Lines 323-324. Lines GeneSwitch- 323 Gal4, S1106-Gal4, GS-Gal4>Aβ42, w 1118 have to be described (here or in Results) and stock numbers should be put in Materials and Methods.

Line 325. Line 43642 should be described.

Lines 330-332. This is not a sentence.

Line 333 What does it mean “tool flies”?

Lines362-363 and 378-379.This is not a protocol. Proper sentences are needed.

Figure S1. The legend is insufficient and incomprehensible. What does it mean “groups”? In which cells are the Gal4 active (see lines 323-324)?

Figure 2. I suggest to put first the expression for the three drivers (A-C) and then lifespan (D-F). In this case, Results should be re-written. Indeed, now 2E is cited before 2D. There
is a problem.

Figure 3D. It is not possible that flies survive up to nine days on 1.5% agarose. They usually die at maximum four days.

Figure 5. Lysotracker is not a measure of lysosomal activity. It is a measure of lysosomal acidity, which is often related to activity, but not always. Measures such as catepsin activity
should be performed.

Author Response

Response to Reviewer 1 Comments

Point 1: Only at the end of the discussion, the authors mention the fact that over-expression of NAGLU cures children with MPS IIIB syndrome (which should be better explained). As a NAGLU-based therapy exists for MPS IIIB patients, what is really missing is a discussion of the interest to use Drosophila to study NAGLU. Also, a better description of the protein function with relevant references is needed.

Response 1: We appreciate this suggestion.

Drosophila as a model system is best suited to study aging and health. Its advantage over mouse model is a tremendous saving of time and cost. The aging study we did in Drosophila lasted approximately 3-4 month, while similar mouse studies take up to 5 years. Even though NAGLU has been studied clinically, it is specifically for the purpose of curing MPS IIIB. Before our study, nothing is known about its potential role in lifespan and other neurodegenerative disease. The interest and justification of using flies to study aging has been dealt with in the introduction, however, as a response, we modified the discussion to enhance this point:

excerpt from the revised discussion:

Studies have reported an intracerebral gene therapy with recombinant AAV vector serotype 2/5 encoding human NAGLU in four children with MPS IIIB syndrome. At intermediate analysis 30 months after surgery, neurocognition was improved in all patients [44]. After 5.5 years of follow-up, good tolerability, sustained NAGLU production, and milder disease were demonstrated in the patient treated at very early stage [45]. As MPS IIIB is also a neurodegenerative lysosomal storage disorder, the improved neurocognition is consistence with what our findings suggest that NAGLU overexpression might alleviate the symptoms of AD. As NAGLU is significantly up-regulated in centenarians, and as its deficiency lead to MPS IIIB, it is essential to investigate its up-regulation is a result of extreme aging, or causal factor for health aging and longevity. Our Drosophila study demonstrated that NAGLU is a positive factor for health aging and longevity and its high expression in centenarians is likely a reason for longevity, rather than a result of extreme aging. Our study also suggests that NAGLU could function as a potential drug target for clinical intervention of aging related neurodegenerative diseases.

Point 2: The manuscript requires professional English editing. Just as an example, at line 27, “overexpressed” should be replaced by ”overexpressing” and a dash should be put (over-expressing).

Response 2: We appreciate this reviewer for pointing this out. We have extensive proof read the text and edited accordingly. The said error has been fixed as below, as throughout the text:

We found that, the deposition of Aβ42 in the mushroom body, the fly central nervous system, was reduced and the lysosomal activity in the intestine was increased in dNAGLU over-expressing flies.

Point 3: Line 44. Aβ is not defined. Should be “amyloid-β (Aβ)” in the sentence before.

Response 3: The revisions have been shown in the Introduction. Also as shown below:

Alzheimer’s disease (AD) is the most common age related disease [2,3], characterized by the accumulation of intracellular neurofibrillary tangles composed of hyperphosphorylated tau protein and the extracellular aggregation of amyloid-β (Aβ) plaques [4-6].

Point 4: Line 63. The Drosophila model of AD should be explained. The manuscript is not written for a non-specialist. Everywhere, its should be Alzheimer’s, not Alzheimer. Reference 26 is not on AD.

Response 4: Thank you so much for your kindly correction, we have edited accordingly, as shown below:

Alzheimer’s disease (AD) is the most common age related disease [2,3], characterized by the accumulation of intracellular neurofibrillary tangles composed of hyperphosphorylated tau protein and the extracellular aggregation of amyloid-β (Aβ) plaques [4-6].

Then we investigated whether and how dNAGLU affected fly life span and health span by overexpressing it in wild type flies and in Drosophila model of AD, which express human Aβ42 and GS-Gal4, simultaneously [26-27].

Point 5: Line 74. Results can not be started with “previous studies”. These should go in the Introduction. Previous data can not be used to make a new figure.

Response 5: The revisions have been shown in the Result section, as shown below:

We collected a group of 117 subjects from centenarian families, consisting of 76 centenarians (CENs) and 41 spouses of centenarian-children (F1SPs, considered as younger adult controls) to analyze the differentially expressed genes in centenarians, and found that NAGLU was up-regulated in CENs (Figure 1A, p = 0.0002), while its expression patterns in controls did not show age related changes (Figure 1B, p = 0.5768).

Point 6: Line 106. “Normally” is inappropriate. A reference should be cited.

Response 6: The revisions have been shown in the Result section, as shown below:

As dNAGLU is expressed at high levels in the fat body and mushroom body [24].

Point 7: Line 127. What is GS-Gal4? In which tissue is it expressed?

Response 7: GS-Gal4 is the GeneSwitch-Gal4 flies, GS-gal4 flies express steroid-activated GAL4 in the whole body of flies. This was discussed when the term first appeared in the text:

Here, we used GeneSwitch (GS), a modified Gal4/UAS inducible expression system, to induce dNAGLU overexpression in specific tissues and its transcriptional activity within the target tissues depends on the presence of the activator mifepristone (RU486) [25]. GS-gal4 flies express steroid-activated GAL4 in the whole body of flies. Then we investigated whether and how dNAGLU affected fly life span and health span by overexpressing it in wild type (wt) flies and in Drosophila model of AD, which express human Aβ42 and GS-Gal4 simultaneously [26-27].

Point 8: Line 128. RU486 is mentioned for the first time and no explanation is given.

Response 8: The modification has been made to the Introduction. Also shown here:

Here, we used GeneSwitch (GS), a modified Gal4/UAS inducible expression system, to induce dNAGLU overexpression in specific tissues and its transcriptional activity within the target tissues depends on the presence of the activator mifepristone (RU486) [25].

Point 9: Lines 127-132. This sentence is too long and incorrect. At least one reference for the AD Drosophila model should be cited.

Response 9: The revisions have been made to the Result section. The details are shown below:

GS-Gal4>Aβ42 flies are homozygous that carries both GS-Gal4 and UAS-Aβ42. Only when RU486 is present, GS-Gal4>Aβ42 flies could induce Aβ42 expression in the whole body. In the control group, the F1 generation obtained by hybridization of GS-Gal4>Aβ42 flies and w1118 flies were fed Sucrose-Yeast (SY) food with 200 μM RU486 to induce ubiquitous Aβ42 expression (i.e. GS-Gal4>Aβ42>+ +RU flies, this is the Drosophila AD model). While in the experimental group, the F1 generation obtained by hybridization of GS-Gal4>Aβ42 flies and dNAGLU transgenic flies were fed SY food with 200 μM RU486 to induce both the expression of Aβ42 and the overexpression of dNAGLU (i.e. GS-Gal4>Aβ42>dNAGLU +RU flies, in other words, we induced dNAGLU overexpression in Drosophila AD model).

Point 10: Line 134. Figure 3 can not be started with G. This should be A.

Response 10 : The orders of the pictures in Figure 3 have been adjusted.

Point 11: Line 135. “Health span” is very vague. Please specify what is meant.

Response 11: “Health span” was described in the Introduction. The original text is as follows:

In AD flies overexpressing human Aβ42, dNAGLU prolongs both life span and health span characterized by fitness and stress resistance test including climbing, desiccation and hunger.

Point 12: Line 179. GS-Gal4>Aβ42>+ -RU is not clear. “+ -” is confusing. Same for “+ +”.

Response 12: GS-Gal4>Aβ42>+ is the expression form of F1 generation flies obtained by  hybridization of GS-Gal4>Aβ42 flies and w1118  flies. The +RU after GS-Gal4>Aβ42>+ means to feed F1 generation SY Food supplemented with RU486. The -RU after GS-Gal4>Aβ42>+ means to feed F1 generation SY Food without RU486 but supplemented with the same dose of ethanol. In the Result 2.3 section, this expression form was explained.

Point 13: Lines 178-183. This sentence is too long and incorrect.

Response 13: The revisions have been shown in the Result. The details are as follows:

We dissected the mushroom bodies of 50-day-old flies for cryo sectioning and performed immunofluorescence staining to observe the deposition of Aβ42 in the brain tissue of these flies.

Point 14: Line 231. A conclusion should be added.

Response 14: The revisions have been shown in the Result section, as shown below:

We then used ELISA to detect the concentration of Aβ42 in cell supernatant, and found that the overexpression led to recovery of cell proliferation (Figure 6B) and the Aβ42 in cell supernatant was reduced by 10.41% (Figure 6C, p = 0.0422). These results demonstrated that when NAGLU was overexpressed in human U251-APP cells, the concentration of Aβ42 in the cell supernatant was reduced, and the growth arrest caused by APP expression was reversed.

Point 15: Line 280. MPS IIIB has to be explained.

Response 15: MPS IIIB was explained in the Introduction, as shown below:

The product of one of these genes, N-acetyl-alpha-glucosaminidase (NAGLU), which degrades heparan sulfate, is an important glucosidase in the lysosomal pathway. Its deficiency leads to clinically termed mucopolysaccharidosis type IIIB (MPS IIIB) [14,15], a devastating and currently incurable neurodegenerative disease, characterized by lysosomal accumulation and urinary excretion of heparan sulfate, and by progressive cognitive decline and behavioral difficulties [16,17]

Point 16: Line 321. What is “ATTP site”?

Response 16: attB/attP is a phiC31 integrase-mediated site-based transgenic system. This system was always used to construct the transgenic flies.

The revision have been made to the Materials and Methods. Also shown below:

The construction of transgenic flies used molecular cloning technology to insert dNAGLU into the UAS promoter-driven vector pUAST-attB-3HA to generate pUAST-attB-3HA-dNAGLU plasmid, which was microinjected into 86F (III) 6110 transgenic strain (gift of Sun Yat-Sen University) with the attP site inserted near 86F.

Point 17: Line 321. What does it mean “dNAGLU transgenic flies can be obtained through mating screening.”? Have they been obtained by standard transgenesis?

Response 17: Yes, dNAGLU transgenic flies was obtained by standard transgenesis. We have corrected the text.

Point 18: Lines 323-324. Lines GeneSwitch-Gal4, S1106-Gal4, GS-Gal4>Aβ42, w1118 have to be described (here or in Results) and stock numbers should be put in Materials and Methods.

Response 18: In the revision, we have added this infomation in the Materials and Methods, and as shown below:

GS-gal4 flies express steroid-activated GAL4 in the whole body. S1106-Gal4 flies express steroid-activated GAL4 in the fat body. 43642 flies express steroid-activated GAL4 in neurons. GS-Gal4>Aβ42 flies express steroid-activated GAL4 and UAS-Aβ42 simultaneously. w1118 flies are wild type flies.

Point 19: Line 325. Line 43642 should be described.

Response 19: 43642 flies express steroid-activated GAL4 in neurons. We have added the description in the revised manuscript.

Point 20: Lines 330-332. This is not a sentence.

Response 20 : Correction has been made in the Materials and Methods, as below:

During the adult experimental stage, flies were fed on Sucrose-Yeast (SY) food (Yeast 100 g/L H2O, Agar 15 g/L H2O, Sugar 50 g/L H2O, Propionic acid 3 mL/L H2O, 10% methyl nuns 30 mL/L H2O).

Point 21: Lines362-363 and 378-379.This is not a protocol. Proper sentences are needed.

Response 20-21: We thank the reviewer for pointing this out. We rewrote the part as shown below:

4.4. Immunofluorescence

The brains of flies were dissected for cryosections, and the brain sections were blocked with blocking buffer (1×PBS/5%BSA/0.3%Triton X-100) at room temperature for 1h. Rinse the blocking buffer with 1×PBS, absorb excess water around the sample with filter paper. The brain sections were incubated with primary antibody: β-Amyloid (CST, 8243S) in the staining buffer (1% BSA and 0.3% Triton X-100 in PBS) at 4°C overnight. The brain sections were washed three times in PBS and incubated with Alexa Fluor 594-conjugated secondary antibodie (ThermoFisher, A-11012) for 2 h at room temperature. Rinse the antibody with 1×PBS, absorb excess water around the sample with filter paper, and mount the slides with Mounting Medium With DAPI (Abcam, ab104139). Imaging was then performed with an Ultra-high-resolution Laser Confocal Microscope (Zeiss, LSM880) and mean fluorescence intensity analysis was performed with Image J.

4.6. qRT-PCR

RNA was extracted with RNAiso Plus (TaKaRa, RNAA00250), then RNA was re-verse transcribed into cDNA with PrimeScript TMRT reagent Kit with gDNA Eraser (TaKaRa, RR047Q), and qRT-PCR was performed by using FastStart Universal SYBR Green Master (ROX) (Roche, 04913850001) and gene-specific primers, all the above experimental operations were performed according to the manufacturer's proce-dures. Gene expression levels were calculated using the comparative CT method. All primers used are listed in (Supplementary Table S1).

Point 22: Line 333 What does it mean “tool flies” ?

Response 22: It means Gal4 flies. Like GS-gal4, S1106-Gal4 and 43642 flies, they are both carry steroid-activated GAL4.

Point 23: Figure S1. The legend is insufficient and incomprehensible. What does it mean “groups”? In which cells are the Gal4 active (see lines 323-324)?

Response 23: Revisions have been made in the Figure S1, dNAGLU transgenic flies were crossed with driver lines GS-gal4, S1106-Gal4 and 43642 flies, respectively. In the experimental group, the F1 generation flies were fed SY Food supplemented with RU486 to induce dNAGLU overexpression in the whole body, fat body and nervous system, respectively. And in the control group, F1 generation flies were fed SY Food without RU486 but supplemented with the same dose of ethanol.

GS-gal4 flies express steroid-activated GAL4 in the whole body. S1106-Gal4 flies express steroid-activated GAL4 in the fat body. 43642 flies express steroid-activated GAL4 in neurons.

Point 24: Figure 2. I suggest to put first the expression for the three drivers (A-C) and then lifespan (D-F). In this case, Results should be re-written. Indeed, now 2E is cited before 2D. There is a problem.

Response 24: We have rewritten the results and make sure that the figures in Fig 2 are described in order, also shown below:

We induced dNAGLUoverexpressionin the fat body and mushroom body, and detected the overexpressed dNAGLU mRNA in the fat body to be 41.52% (p = 0.0002) higher than that in the control group (Figure 2C), and the expression of 3HA-dNAGLU protein was detected by Western blot (Supplementary FigureS2). Statistical analysis of lifespan showed that the overexpression of dNAGLU in the fat body prolonged the lifespan of Drosophila, and mean lifespan increased by 2.19% (Figure 2D, p = 0.0008). When dNAGLU was knocked down in the Drosophila fat body,the lifespan was significantly decreased in comparison with the control group(Supplementary FigureS3). In the CNS, dNAGLU mRNA expression was higher than the control group (Figure 2E), with an increase of 177.9% (p = 0.0010). And the overexpression of dNAGLUin the nervous system prolonged the lifespan of Drosophila, and mean lifespan increased by 2.55% (Figure 2F, p = 0.0195).

Point 25: Figure 3D. It is not possible that flies survive up to nine days on 1.5% agarose. They usually die at maximum four days.

Response 25: Flies with different treatments have different responses to stress. For the starvation test, flies survive up to nine days on 1.5% agarose is not rare. For example, there is a research in PNAS, they used 1% agarose for starvation test in flies, their results showed that flies survive up to 13 days (Broughton SJ et al, 2005). Another research in Aging-US, they used 1.5% agarose for starvation test in flies, their results showed that flies survive up to 15 days (Fan X et al, 2021).

Point 26: Figure 5. Lysotracker is not a measure of lysosomal activity. It is a measure of lysosomal acidity, which is often related to activity, but not always. Measures such as catepsin activity should be performed.

Response 26: Some articles mention that lyso tracker red can be used as an indicator of lysosomal biogenesis (Yin Q et al.,2020). And in Cold Spring Harbor Protocols, it describes: LysoTracker is an acidotropic dye that stains cellular acidic compartments, including lysosomes and autolysosomes. LysoTracker has been used to detect autophagy-associated lysosomal activity in Drosophila tissues including the fat body, midgut, salivary gland and ovary, as well as in Drosophila cell culture (DeVorkin L and Gorski SM, 2014). In addition, in figure 6H, we detect the transcript levels of the Cts family of cysteine proteases: i.e. CTS-B, -D, -K and -L and TFEB /TFE3. We find that when NAGLU overexpressed in U251-APP cells, transcript levels of CTSB, CTSD, CTSK, CTSL, TPP1, TFEB and TFE3 were significantly increased. It exhibited strong lysosomal activity and promoted lysosomal biogenesis. As dNAGLU and NAGLU are homologous, they are usually functionally conservative. Taken together, our results of the enhanced fluorescence intensity of lyso tracker staining is related to a strong lysosomal activity.

References:

Broughton SJ, Piper MD, Ikeya T, Bass TM, Jacobson J, Driege Y, Martinez P, Hafen E, Withers DJ, Leevers SJ, Partridge L. Longer lifespan, altered metabolism, and stress resistance in Drosophila from ablation of cells making insulin-like ligands. Proc Natl Acad Sci U S A. 2005 Feb 22;102(8):3105-10. doi: 10.1073/pnas.0405775102. Epub 2005 Feb 11. PMID: 15708981; PMCID: PMC549445.

DeVorkin L, Gorski SM. LysoTracker staining to aid in monitoring autophagy in Drosophila. Cold Spring Harb Protoc. 2014 Sep 2;2014(9):951-8. doi: 10.1101/pdb.prot080325. PMID: 25183815.

Fan X, Zeng Y, Fan Z, Cui L, Song W, Wu Q, Gao Y, Yang D, Mao X, Zeng B, Zhang M, Ni Q, Li Y, Wang T, Li D, Yang M. Dihydromyricetin promotes longevity and activates the transcription factors FOXO and AOP in Drosophila. Aging (Albany NY). 2020 Dec 3;13(1):460-476. doi: 10.18632/aging.202156. Epub 2020 Dec 3. PMID: 33291074; PMCID: PMC7835053.

Yin Q, Jian Y, Xu M, Huang X, Wang N, Liu Z, Li Q, Li J, Zhou H, Xu L, Wang Y, Yang C. CDK4/6 regulate lysosome biogenesis through TFEB/TFE3. J Cell Biol. 2020 Aug 3;219(8):e201911036. doi: 10.1083/jcb.201911036. PMID: 32662822; PMCID: PMC7401801.

Thanks very much for taking your time to review this manuscript. I really appreciate all your comments and suggestions!

Reviewer 2 Report

The manuscript by Xue, et al., reports that overexpression of the gene dNAGLU in Drosophila extends lifespan, enhances lysosome biogenesis, and suppresses accumulation of expressed human Abeta, while expression of the human ortholog in human cultured cells reduces accumulation of secreted Abeta in the medium. It is difficult to know how to interpret these findings. Each individual experiment is reported as having a statistically significant result in the suggested direction. However, the magnitudes of most of these effects are tiny, and one can’t help being skeptical about whether they would reproduce if repeated with independent data collections. Moreover, key controls are lacking for a number of these experiments. Therefore, while there is nothing reported in the manuscript that actively argues against the authors’ interpretations, I would need to see replication of a number of the experiments to be convinced of the robustness of the conclusions of the study. Moreover, perhaps a deeper problem is that even if I accept the results at face value, while they do show effects of dNAGLU on various organismal measures, like lifespan, and on autophagy, they do not show that the autophagy effects are *causal* for the lifespan effects, which is a major conclusion of the manuscript.

Details are below.

1.     The main problem with the paper is that the small magnitude of the observed effects makes one wonder whether they are real, and whether they reflect direct effects of the targeted manipulation or unrelated covariates. This is certainly true for the lifespan effects (Fig 2) as well as the effects on physiology (Fig 3). For example, the authors do a control of leaving out the inducer, RU, but they do not do a control providing RU but leaving out the Geneswitch GAL4 element or the responding UAS transgene. Perhaps RU itself has a very small but reproducible effect on lifespan. 

2.     Related to #1, there are many factors that alter lifespan, particularly in females (as used in this study), including fecundity and food consumption. Did the authors verify that food intake and fecundity were not altered by RU and/or transgene expression? Given the small magnitude of the effects, even very mild effects on these other processes could easily account for the observed outcomes. Similarly, small effects of RU on the composition of the microbiome could easily produce lifespan effects of the observed magnitude.

3.     Also related to #1, the lifespan effects are on the order of 0.1% - 3.6%. Lifespan is notoriously irreproducible from trial to trial. It is hard to believe that an effect this small is robust. The authors state that they used several vials of flies in parallel for each measurement, which is good, but did they repeat the entire lifespan studies, start to finish, with that number of flies multiple times, and if so, was the effect of dNAGLU expression quantitatively the same in each independent trial? 

4.     Measurements of survival of Abeta-expressing flies, starvation stress, and dessication stress (Fig 3) share the same problems of small effect size and potential confounders as do the lifespan experiments.

5.     The authors report the effect of NAGLU overexpression. Did they investigate lifespan of a heterozygous or homozygous mutant of dNAGLU, or of providing a NAGLU inhibitor? Seeing the opposite effect in the loss-of-function condition would greatly strengthen the interpretations.

6.     The fact that massive overexpression of NAGLU (100x) produces such modest change in the level of secreted Abeta (~10%), is one of the observations that forces one to question whether the biochemical effects the authors are focusing on is really the source of the organismal effects that they cite.

7.     Even if I accept the results of individual experiments, the authors do not have an experiment that demonstrates causality. They conclude that the effects of dNAGLU on autophagy are responsible for the observed effects on stress sensitivity and lifespan. However, they do not do any experiment that selectively blocks the effects of dNAGLU overexpression on autophagy and shows that this blocks the organismal outcomes. dNAGLU may also be doing something entirely unrelated to autophagy that is the pathway affecting lifespan. Without some experiment to demonstrate necessity, the fundamental conclusion cannot be supported.

Minor details.

A.    In Fig 1, the units of NAGLU expression in panels A and B are entirely different from those in panel C. It would be preferable to use the same units, or at least to explain the relationship between the two.

Author Response

Response to Reviewer 2 Comments

Point 1. The main problem with the paper is that the small magnitude of the observed effects makes one wonder whether they are real, and whether they reflect direct effects of the targeted manipulation or unrelated covariates. This is certainly true for the lifespan effects (Fig 2) as well as the effects on physiology (Fig 3). For example, the authors do a control of leaving out the inducer, RU, but they do not do a control providing RU but leaving out the Geneswitch GAL4 element or the responding UAS transgene. Perhaps RU itself has a very small but reproducible effect on lifespan.

Response 1:

First of all, human longevity is a combination of multiple positive factors, it is unlikely any individual factor would have a huge positive impact on lifespan. We should not expect such in animal experiments for factors identified from human longevity individuals. That said, the referred lifespan experiments were done three times, independently, by the two authors, over a period of more than one year. We are certain that these results are repeatable. In response to your question, the additional lifespan results not included in the previous manuscript have been added to the supplemental data (Fig. S4 and S5).

RU486 used in this study is widely used in the field of aging (Alic et al., 2014; Dobson et al., 2019; Hwangbo et al., 2004), and there should not be any doubt that it alone has any significant effect on lifespan. We verified the effect of RU486 on the life span and fecundity, and showed that at the concentration of 200 μM, there was no significant difference in the lifespan (Fig. S1B) or in the fecundity of fruit flies (Fig.S1A). Consequently, the concentration of 200 μM of RU486 was used in our experiments. Since the results (Fig. S1B) showed that RU486 at 200 μM does not affect the lifespan of fruit flies, we therefore conclude that the overexpression of NAGLU was the cause of the increased lifespan of fruit flies.

The referenced figures are included below:

Supplementary Figure S1. Effects of different concentration of RU486 on the fecundity and lifespan in Drosophila. (A) Fecundity evaluation of w1118 flies, p = 0.0504 (B) Survival curves generated from life statistics of w1118 flies , p = 0.3471. ns represents non-significant. All survival data were analyzed with the log-rank test. The experiment was repeated for three times.

Supplementary Figure S4. Repeated experiment of dNAGLU overexpression on the lifespan in Drosophila. (A) qRT-PCR validation for expression of dNAGLU in Drosophila whole body. (B) Survival curves of overexpressing dNAGLU in Drosophila whole body, P = 0.0235. (C) qRT-PCR validation for expression of dNAGLU in Drosophila fat body. (D) Survival curves of overexpressing dNAGLU in Drosophila fat body, p = 0.0244. (E) qRT-PCR validation for expression of dNAGLU in Drosophila nervours system. (F) Survival curves of overexpressing dNAGLU in Drosophila nervours system, P = 0.0259. A 5% or lower P value is considered to be statistically significant using Student’s t-test, *p<0.05, ** p<0.01, *** p<0.001. All survival data were analyzed with the log-rank test. The experiment was repeated for three times.

Supplementary Figure S5. Repeated experiment of dNAGLU overexpression on the lifespan, fitness and tress resistance in Drosophila AD models. (A) qRT-PCR validation for expression of dNAGLU in AD flies (** p<0.01, t-test ). (B) Survival curve of Drosophila overexpressing dNAGLU in AD flies (p = 0.0196, log-rank test). (C-F) Effects of dNAGLU overexpression on stress tolerance in AD flies. (C) Climbing assay, (*p<0.05, t-test). (D) Hunger stress assay, (p = 0.0003, log-rank test). (E) Desiccation stress assay, (p = 0.0008, log-rank test). (F) H2O2 stress assay, (p = 0.1130, log-rank test). The experiment was repeated for three times.

Point 2. Related to #1, there are many factors that alter lifespan, particularly in females (as used in this study), including fecundity and food consumption. Did the authors verify that food intake and fecundity were not altered by RU and/or transgene expression? Given the small magnitude of the effects, even very mild effects on these other processes could easily account for the observed outcomes. Similarly, small effects of RU on the composition of the microbiome could easily produce lifespan effects of the observed magnitude.

Response 2:We agree that these two are essential controls.

First, we found that at 200 μM concentration of RU486 used did not affect fecundity in w1118 flies used in the experiments (Fig. S1A). Neither did we see any changes in fecundity after overexpression of dNAGLU (Fig. S8AB).

Second, a control on the effects on food intake by different concentration of RU486 were done

in a separate, but related study conducted in our lab, at 200 μM, there was no effect on food intake (Fig. S8C).

For these reasons, our conclusion of NAGLU on lifespan was well supported.

Figure S8. Effects of dNAGLU overexpression in Drosophila. (A) Eggs laying tendency of overexpressing dNAGLU in Drosophila whole body, p = 0.1533. (B) Eggs laying tendency of overexpressing dNAGLU in Drosophila fat body, p = 0.1417. (C) Assessment of food intake in RU486 induced system, p = 0.8772.

Point 3. Also related to #1, the lifespan effects are on the order of 0.1% - 3.6%. Lifespan is notoriously irreproducible from trial to trial. It is hard to believe that an effect this small is robust. The authors state that they used several vials of flies in parallel for each measurement, which is good, but did they repeat the entire lifespan studies, start to finish, with that number of flies multiple times, and if so, was the effect of dNAGLU expression quantitatively the same in each independent trial?

Response 3: First of all, like we stressed earlier, human longevity is a combination of multiple positive factors, it is unlikely any individual factor would have a huge positive impact on lifespan. We should not expect such in animal experiments when we studying potential human longevity factors. That said, the referred lifespan experiments were done three times, independently, by the two authors, over a period of more than one year. We are certain that these results are repeatable. In response to your question, the additional lifespan results not included in the previous manuscript have been added to the supplemental data (Fig. S4, S5, also see above).

Point 4. Measurements of survival of Abeta-expressing flies, starvation stress, and dessication stress (Fig 3) share the same problems of small effect size and potential confounders as do the lifespan experiments.

Response 4:Similar to lifespan studies, these experiments were repeated independently by different authors of the manuscript, and similar results were obtained, although we only included on set of the experimental results in the original manuscript. In response, we included a repeat experiment in the revised manuscript as supplementary data (Fig. S5, also see above). We are certain from these experiments that overexpression of dNAGLU in AD flies extends the lifespan, improves muscle strength, and the resistance to desiccation and hunger.

Point 5. The authors report the effect of NAGLU overexpression. Did they investigate lifespan of a heterozygous or homozygous mutant of dNAGLU, or of providing a NAGLU inhibitor? Seeing the opposite effect in the loss-of-function condition would greatly strengthen the interpretations.

Response 5:There is no NAGLU inhibitor, neither were there dNAGLU mutant flies. However, we did perform RNAi knockdown of dNAGLU in flies, both in the whole body and in fat body. As expected, we observed that the knockdown significantly shortens lifespan. In response, we included these results in the supplementary data in this revision (Figure S3 also see below).

Supplementary Figure S3. Effects of dNAGLU knockdown on the lifespan in Drosophila. (A) qRT-PCR validation for expression of dNAGLU in Drosophila whole body. (B) Survival curves when dNAGLU was knocked down in Drosophila whole body, P = 0.0374. (C) qRT-PCR validation for expression of dNAGLU in Drosophila fat body. (D) Survival curves when dNAGLU was knocked down in Drosophila fat body, P = 0.0392. ** p<0.01, *** p<0.001. All survival data were analyzed with the log-rank test. The experiment was repeated for three times.

Point 6. The fact that massive overexpression of NAGLU (100x) produces such modest change in the level of secreted Abeta (~10%), is one of the observations that forces one to question whether the biochemical effects the authors are focusing on is really the source of the organismal effects that they cite.

Response 6:The 100 fold increase seen in Figure 6 (included below) reflected the low mRNA levels in these cells, and not necessarily indicate “massive amount” of the NAGLU protein. As could be seen in the Western blot analysis in the supplemental Fig. S7AB, the level of extrinsic, induced NAGLU protein (FLAG-NAGLU fusion protein) is comparable to that of the APP protein these cells were also expressing.

We included this data in the revised manuscript as a supplement (Fig. S7AB, also see below).

Figure 6. Effects of NAGLU overexpression in human U251-APP cells. (A) qRT-PCR validation for expression of NAGLU in U251-APP cells ( *** p<0.001, t-test ). (B) Cell proliferation curve for NAGLU overexpressed cells detected by cell counting assay (*p<0.05, ** p<0.01, t-test ). (C) Detection of Aβ42 concentration in cell supernatants by ELISA (*p<0.05,t-test).

Supplementary Figure S7. Western Blot detection when NAGLU overexpressed in U251-APP cells. (A) Western Blot detection of Flag-NAGLU, APP and β-actin. (B) IOD value of APP and β-actin, ns represents non-significant.

Point 7. Even if I accept the results of individual experiments, the authors do not have an experiment that demonstrates causality. They conclude that the effects of dNAGLU on autophagy are responsible for the observed effects on stress sensitivity and lifespan. However, they do not do any experiment that selectively blocks the effects of dNAGLU overexpression on autophagy and shows that this blocks the organismal outcomes. dNAGLU may also be doing something entirely unrelated to autophagy that is the pathway affecting lifespan. Without some experiment to demonstrate necessity, the fundamental conclusion cannot be supported.

Response 7: We respectably disagree. First of all, we did clearly established a causal relationship between dNAGLU, homologue of NAGLU, an overexpressed lysosomal enzyme in centenarians, and the lifespan and health-span extension in flies:

First, we revealed that NAGLU could extend life span (Fig. 2) and health span in flies (Fig. 3), and enhance β amyloid clearance in AD flies (Fig. 4) and in AD cells (Fig. 6C). Based on the biological function of NAGLU, we demonstrated by lyso-traker staining that the overexpression of dNAGLU increased lysosomal activity in both AD flies (Fig. 5) and AD cells (Fig. 6DEFG). Enhancement of lysosomal function clearly promoted the clearance of β amyloid.

Second, and more importantly, we found that the overexpression of NAGLU in AD cells increased the expression of genes associated with β amyloid clearance, in particular TFEB and TFE3, two genes that promote lysosomal gene expression, increased significantly (Fig. 6H). These results suggest that the overexpression of NAGLU enhanced lysosomal function, and thus enhanced β amyloid clearance, which served as a mechanistic basis of lifespan extension, stress resistance and improved fitness in overexpressed flies.

We appreciate this reviewer for proposing additional experiments that might further prove the causal effect. However, there is no specific inhibitor for NAGLU, and commonly used lysosomal inhibitors could not specifically block the effects of dNAGLU overexpression without introducing additional, complicating side effects. For example, leupeptin only inhibits proteases, while chloroquine changes is the pH in lysosomes. These inhibitors will certainly disrupt more, additional biological function of the cell than you would imagine.

In theory, specific inhibitor for, or a catalytic site point mutation in NAGLU might achieve the proposed function, but inhibiting NAGLU would have a similar, or even stronger effects than NAGLU knockdown, which is known to shorten lifespan (supplemental figure S3), not to mention the deficiency of this enzyme is known to lead to human disease called MPS IIIB. The alternative, overexpressing a dead enzyme would also affect endogenous NAGLU activity by competing for substrates, or cofactors, which will similarly shorten lifespan as discussed above, and as seen in the newly included supplemental figure S3 (also see above).

We appreciate any further suggestions of viable experiments that could prove that NAGLU’s enzymatic activity, or any other activities it may have, is linked to its role in lifespan and health-span extension.

Minor details.

Point 8. In Fig 1, the units of NAGLU expression in panels A and B are entirely different from those in panel C. It would be preferable to use the same units, or at least to explain the relationship between the two.

Response 8: Fig 1A and 1B are analyzed from blood samples in centenarians (CENs) and spouses of centenarian-children (F1SPs). The DESeq2 package was used to identify the differentially expressed genes between CENs and F1SPs, vst transformed read counts by “vst” function in DESeq2 package were used to represent the genes expression levels. Fig. 1C is analyzed from the dataset of genome-wide RNA-seq profiles of human dermal fibroblasts from 133 people aged 1 to 94 years old (Fleischer JG et al, 2018). FPKM transformed read counts were used to represent the genes expression levels. Gene expression is inherently a relative quantity and gene expression varies from sample to sample. We used internal comparisons of samples from the same batch and did not involve comparisons between samples from different batches. So, it is reasonable to use different transform method, which can reflect our gene expression more accurately. Fig 1A and 1B were analyzed by vst transformed read counts, Fig. 1C was analyzed by FPKM transformed read count. Therefore, their gene expression units are different.

References

Alic, N., Giannakou, M.E., Papatheodorou, I., Hoddinott, M.P., Andrews, T.D., Bolukbasi, E., and Partridge, L. (2014). Interplay of dFOXO and two ETS-family transcription factors determines lifespan in Drosophila melanogaster. PLoS Genet 10, e1004619. 10.1371/journal.pgen.1004619.

Dobson, A.J., Boulton-McDonald, R., Houchou, L., Svermova, T., Ren, Z., Subrini, J., Vazquez-Prada, M., Hoti, M., Rodriguez-Lopez, M., Ibrahim, R., et al. (2019). Longevity is determined by ETS transcription factors in multiple tissues and diverse species. PLoS Genet 15, e1008212. 10.1371/journal.pgen.1008212.

Fleischer JG, Schulte R, Tsai HH, Tyagi S, Ibarra A, Shokhirev MN, Huang L, Hetzer MW, Navlakha S. Predicting age from the transcriptome of human dermal fibroblasts. Genome Biol. 2018 Dec 20;19(1):221. doi: 10.1186/s13059-018-1599-6.

Hwangbo, D.S., Gersham, B., Tu, M.-P., Palmer, M., and Tatar, M. (2004). Drosophila dFOXO controls lifespan and regulates insulin signalling in brain and fat body. Nature 429, 562-566. 10.1038/nature02549.

Round 2

Reviewer 1 Report

See attached file.

Author Response

Firstly, we thank this reviewer to painstakingly helping us identify the numerous errors in the manuscript, greatly appreciated!

Point 1: - the improved neurocognition is consistentce with what our findings suggest…

Response 1: Thank you so much for your kindly correction, the revised text is shown below:

As MPS IIIB is also a neurodegenerative lysosomal storage disorder, the improved neurocognition is consistent with what our findings suggest that NAGLU overexpression might alleviate the symptoms of AD.

Point 2: GS-gal4 should be always GS-Gal4 and Gal4 always capitalized.

Response 2: We have made corrections in the text. Now all Gal4 of GS-Gal4 are capitalized.

Point 3: GS-Gal4>Aβ42 flies are homozygous that carriescarry both GS-Gal4 and UAS-Aβ42.

Response 3: We have edited as shown below:

GS-Gal4>Aβ42 flies are homozygous that carry both GS-Gal4 and UAS-Aβ42.

Point 4: GS-gal4 flies -we used express steroid-activated GAL4 in the whole body of flies. Then

we investigated whether and how dNAGLU affected fly life span and health span by overexpressing it in wild type (wt) flies and in the Drosophila model of AD, which expresses human Aβ42 and GS-Gal4 simultaneously [26-27].

Response 4: Thanks for your kindly correction, we have edited accordingly, as shown below:

The GS-Gal4 flies we used express steroid-activated GAL4 in the whole body. Then we investigated whether and how dNAGLU affected fly life span and health span by overexpressing it in wild type (wt) fliesand in the Drosophila model of AD, which expresses human Aβ42 and GS-Gal4 simultaneously [26-27].

Point 5: We tested the effects of RU486 on lifespan and fecundity and findedfound that 200 µM RU486 had no significant effect on lifespan and fecundity of wt flies (Supplementary Figure S1). So we used 200 µM RU486 to induce dNAGLU overexpression. (delete dash) We overexpressed dNAGLU...

Response 5: The revisions have been shown in the text, as shown below:

We tested the effects ofRU486 on lifespan and fecundity and found that 200 μM RU486 had no significant effect on lifespan and fecundity of wt flies(Supplementary FigureS1). So we used 200 μM RU486 to induce dNAGLU overexpression. We overexpressed dNAGLU ubiquitously and observed its effects on Drosophila lifespan.

Point 6: Line 225: the sentence “While we used GS-Gal4>Aβ42>dNAGLU +RU flies (Figure 4C),

which induce the overexpression of dNAGLU in Drosophila AD model, as the

experimental group. “can not be started with while.

Response 6: The revisions have been shown in the text, as shown below:

We used GS-Gal4>Aβ42>dNAGLU +RU flies (Figure 4C), which induce the overexpression of dNAGLU in the Drosophila AD model, as the experimental group.

Point 7: Line 273: eliminate comma.

Response 7: The revisions are shown as below:

These results demonstrated that when NAGLU was overexpressed in human U251-APP cells, the concentration of Aβ42 in the cell supernatant was reduced and the growth arrest caused by APP expression was reversed.

Point 8: dNAGLU transgenic flies were crossed with driver lines GS-gal4 (added spaces),

Response 8: The revisions are shown as below:

dNAGLU transgenic flies were crossed with driver lines GS-Gal4, S1106-Gal4 and 43642 flies, respectively.

Point 9: The Immunofluorescence paragraph is still written as a protocol (Rinse the blocking buffer…, Rinse the antibody…)

Response 9: We have made revisions to make it reads more like a description than a protocol, as shown below:

4.4. Immunofluorescence

The brains of flies were dissected for cryosections, and the brain sections were blocked with blocking buffer (1×PBS/5%BSA/0.3%Triton X-100) at room temperature for 1hour. The slides were incubated with primary antibody: β-Amyloid (CST, 8243S) in the staining buffer (1% BSA and 0.3% Triton X-100 in PBS) at 4°C overnight and subsequently incubated with Alexa Fluor 594-conjugated secondary antibody (ThermoFisher, A-11012) for 2 hours at room temperature. Mount the slides with Mounting Medium With DAPI (Abcam, ab104139). Imaging was then performed with an Ultra-high-resolution Laser Confocal Microscope (Zeiss, LSM880) and mean fluorescence intensity analysis was performed with Image J.

Point 10: “all the above experimental operations were performed according to the manufacturer's

Procedures” should be preceded by a period, not a comma.

Response 10 : The revisions are shown as below:

qRT-PCR was performed using the Fast Start Universal SYBR Green Master (ROX) (Roche, 04913850001) and gene-specific primers. All the above experimental operations were performed according to the manufacturer's procedures.

Thanks very much for taking your time to review this manuscript. I really appreciate all your comments and suggestions! 

Reviewer 2 Report

The authors have sensible responses to most of my central objections to the manuscript. My main concern was that they provide evidence that the small effects they see on lifespan and stress survival are (a) reproducible, and (b) not unrelated effects of RU activation of the various GS elements they. It is extremely helpful that they now show data for a complete, independent repeat of each of the lifespan and stress-resistance experiments. There are a couple of additional minor points that it would be well to clarify.

1.     The authors state that they have a third, independent trial for each lifespan and stress survival experiment. It would be sensible also to include a supplemental spreadsheet containing the raw data for this additional trial, for the benefit of any reader who would like to see the entire set.

2.     My copy of the supplemental files did not include supplement S8, which includes a few of the controls I requested. I do not question the authors’ description of those results, but they should be sure to provide a complete supplemental file with the final manuscript.

3.     Regarding the question of whether RU activation in a geneswitch-bearing fly could have a small effect lifespan independent of the downstream UAS-dNAGLU, adding RU without any GS element is certainly not a perfect control; GAL4 itself has significant effects on global gene expression even in the absence of any UAS element. The control with GS but without the UAS line, or with an unrelated control UAS-line, would have been better. However, the authors’ evidence that UAS-RNAi has a lifespan effect in the opposite direction from that of UAS-dNAGLU ameliorates that concern. I therefore do not insist on that control being added. 

4.     Regarding the question of causality, the issue is not whether dNAGLU affects autophagy, nor (given the data now provided) that it affects survival. The question is whether it hs been established that the autophagy effect is causal for the lifespan effect in this experimental paradigm. I recognize the technical challenges in doing this, and I do not question whether, in general, it is plausible to suggest that autophagy is a natural candidate for the link between dNAGLU and lifespan. However, it remains the case that the causality of this specific interaction has not been demonstrated. A simple sentence in the Discussion pointing out this caveat would be sufficient.

5.     Regarding loss of function reagents for dNAGLU, Flybase identifies a known null allele of dNAGLU that is available from the Bloomington Drosophila Stock Center. I do not insist that the authors hold up the paper while they perform experiments with this allele (there are many reasons why that could be problematic, and it would certainly be a lengthy experiment), but I would encourage the authors to consider testing this reagent in the future.

Author Response

Point 1. The authors state that they have a third, independent trial for each lifespan and stress survival experiment. It would be sensible also to include a supplemental spreadsheet containing the raw data for this additional trial, for the benefit of any reader who would like to see the entire set.

Response 1:Thanks for your suggestions. We have added the raw data of three independent repeated experiments of life span and health span to the supplemental materials, and the raw data was contained in the pzfx format file.

Point 2. My copy of the supplemental files did not include supplement S8, which includes a few of the controls I requested. I do not question the authors’ description of those results, but they should be sure to provide a complete supplemental file with the final manuscript.    

Response 2:Thanks for your corrections. We have made some revisions in the supplemental file. The supplementary materials we uploaded this time are complete.

Point 3. Regarding the question of whether RU activation in a geneswitch-bearing fly could have a small effect lifespan independent of the downstream UAS-dNAGLU, adding RU without any GS element is certainly not a perfect control; GAL4 itself has significant effects on global gene expression even in the absence of any UAS element. The control with GS but without the UAS line, or with an unrelated control UAS-line, would have been better. However, the authors’ evidence that UAS-RNAi has a lifespan effect in the opposite direction from that of UAS-dNAGLU ameliorates that concern. I therefore do not insist on that control being added.

Response 3: Thanks for your suggestions. In theory, any perturbation could have a potential consequences in lifespan and health span, and being careful in designing experimental controls is always a good idea. However, since 1993 (Brand AH and Perrimon N,1993), when GAL4/UAS system was introduced, thousands of Gal4 driver lines that have been developed (Busson D and Pret AM, 2007; Caygill EE and Brand AH, 2016; di Pietro F et al., 2021), and used in all areas of Drosophila research including aging studies. For this reason, and subsequent results from our UAS-RNAi experiment, we did not include the suggested control. We will, however, consider similar strict controls in the future. We thanks this reviewer for not insisting.

Point 4. Regarding the question of causality, the issue is not whether dNAGLU affects autophagy, nor (given the data now provided) that it affects survival. The question is whether it hs been established that the autophagy effect is causal for the lifespan effect in this experimental paradigm. I recognize the technical challenges in doing this, and I do not question whether, in general, it is plausible to suggest that autophagy is a natural candidate for the link between dNAGLU and lifespan. However, it remains the case that the causality of this specific interaction has not been demonstrated. A simple sentence in the Discussion pointing out this caveat would be sufficient.

Response 4:Now we completely understood the original question, and we agree, that what (the detailed mechanism) mediates the lifespan and health span effect of NAGLU is not understood. As mentioned in the discussion, it could be an effect not related to its catalytic activity, but that is beyond the scope of this study and needs significant further investigation. Our current study to report that NAGLU has functions in addition to its catalytic activity to promote health and lifespan. Its overexpression in centenarians could be a reason for longevity, rather than a consequence of aging.

In response to your comments, we modified the discussion as shown below:

Thus, NAGLU may promote lysosomal biogenesis through up-regulating TFEB/TFE3, which increase the expression of a number of proteolytic enzymes that degrade Aβ in lysosomes, and facilitate the degradation of Aβ42 (Figure 7). The mechanism leading to up-regulation of TFEB/TFE3 is currently unknown and the relationship between NAGLU overexpression and autophagy also needs to be explored. These are interesting questions for future studies.

Point 5. Regarding loss of function reagents for dNAGLU, Flybase identifies a known null allele of dNAGLU that is available from the Bloomington Drosophila Stock Center. I do not insist that the authors hold up the paper while they perform experiments with this allele (there are many reasons why that could be problematic, and it would certainly be a lengthy experiment), but I would encourage the authors to consider testing this reagent in the future.

Response 5:Thanks for your kindly suggestions. We will continue our studies on NAGLU, and the fly strain you mentioned will be obtained and included in future studies.

References:

Brand AH, Perrimon N. Targeted gene expression as a means of altering cell fates and generating dominant phenotypes. Development. 1993 Jun;118(2):401-15. doi: 10.1242/dev.118.2.401. PMID: 8223268.

Busson D, Pret AM. GAL4/UAS targeted gene expression for studying Drosophila Hedgehog signaling. Methods Mol Biol. 2007;397:161-201. doi: 10.1007/978-1-59745-516-9_13. PMID: 18025721.

Caygill EE, Brand AH. The GAL4 System: A Versatile System for the Manipulation and Analysis of Gene Expression. Methods Mol Biol. 2016;1478:33-52. doi: 10.1007/978-1-4939-6371-3_2. Erratum in: Methods Mol Biol. 2016;1478:E1-E3. PMID: 27730574.

di Pietro F, Herszterg S, Huang A, Bosveld F, Alexandre C, Sancéré L, Pelletier S, Joudat A, Kapoor V, Vincent JP, Bellaïche Y. Rapid and robust optogenetic control of gene expression in Drosophila. Dev Cell. 2021 Dec 20;56(24):3393-3404.e7. doi: 10.1016/j.devcel.2021.11.016. Epub 2021 Dec 7. PMID: 34879263; PMCID: PMC8693864.